# Ovarian Endometrioid and Clear Cell Carcinomas with Low Prevalence of Microsatellite Instability: A Unique Subset of Ovarian Carcinomas Could Benefit from Combination Therapy with Immune Checkpoint Inhibitors and Other Anticancer Agents

**DOI:** 10.3390/healthcare10040694

**Published:** 2022-04-07

**Authors:** Yuki Nonomura, Kentaro Nakayama, Kohei Nakamura, Sultana Razia, Hitomi Yamashita, Tomoka Ishibashi, Masako Ishikawa, Seiya Sato, Satoru Nakayama, Yoshiro Otsuki, Satoru Kyo

**Affiliations:** 1Department of Obstetrics and Gynecology, Shimane University School of Medicine, Izumo 6938501, Japan; y-yuki@med.shimane-u.ac.jp (Y.N.); knakamura320@keio.jp (K.N.); raeedahmed@yahoo.com (S.R.); memedasudasu1103@gmail.com (H.Y.); tomoka@med.shimane-u.ac.jp (T.I.); m-ishi@med.shimane-u.ac.jp (M.I.); sato_seiya9534@yahoo.co.jp (S.S.); satoruky@med.shimane-u.ac.jp (S.K.); 2Department of Obstetrics and Gynecology, Seirei Hamamatsu Hospital, Hamamatsu 4308558, Japan; satoru@sis.seirei.or.jp; 3Department of Organ Pathology, Seirei Hamamatsu Hospital, Hamamatsu 4308558, Japan; otsuki@sis.seirei.or.jp

**Keywords:** ovarian endometrioid carcinoma, ovarian clear cell carcinoma, microsatellite instability, immune checkpoint inhibitors

## Abstract

Ovarian cancer has the highest mortality rate among all gynecological malignancies; therefore, a novel treatment strategy is needed urgently. Utilizing immune checkpoint inhibitors has been considered for microsatellite instability (MSI)-high (MSI-H) tumors. However, the prevalence of MSI-H tumors in ovarian endometrioid and clear cell carcinomas remains unclear. Here, polymerase chain reaction was used to analyze 91 cases of ovarian endometrioid and clear cell carcinomas for the MSI status and the relationship between MSI-H, immune checkpoint molecules, and clinicopathological factors (including patient survival). Only 5 of 91 (5%) cases were MSI-H endometrioid carcinomas. In these cases, CD-8 expression was significantly higher (*p* = 0.026), confirming an enhanced immune response. From the survival curve, no statistical correlations were found between the MSI-H group and the microsatellite stable (MSS) group; however, the MSS group trended towards better progression-free survival than the MSI-H group (*p* = 0.056). Patients with PD-L1 expression had shorter overall survival than those without (*p* = 0.022). Thus, MSI-H is a rare event and not a favorable prognostic factor in ovarian endometrioid and clear cell carcinomas. Thus, to improve the prognosis of ovarian endometrioid carcinoma and clear cell carcinomas, a combination therapy of immune checkpoint inhibitors and other molecular targeted therapies may be required.

## 1. Introduction

Globally, in 2020, there were more than 300,000 new of cases ovarian cancer which resulted in more than 190,000 deaths [1]. The 5-year survival rate for ovarian cancer is 60%, which is a poor prognosis, and it has the highest mortality rate among gynecological cancers [2,3]. Thus, there is an urgent need to discover novel treatments for ovarian cancer. Recently, due to the development of genomic medicine, molecular-targeted drugs have attracted attention. There have been reports that immune checkpoint inhibitors may be effective in microsatellite instability (MSI)-high (MSI-H) tumors [4]. MSI is a unique molecular change and hypermutable phenotype caused by a defective DNA mismatch repair (MMR) system [5]. In Japan, immune checkpoint inhibitors for MSI-H tumors were covered by insurance in 2018.

A mismatch repair (MMR) is a DNA repair mechanism that repairs the damage caused by DNA replication errors. However, when the function of MMR proteins (MSH2, MSH6, PMS2, and MLH1) is impaired due to mutations or methylation, the DNA replication error due to the mismatch cannot be repaired and the gene abnormalities accumulate [6]. In addition, gene mutations are likely to occur in microsatellite lesions during DNA replication. An MMR deficiency cannot repair the error, which may result in an abnormal number of iterations in the microsatellite lesion. Recently, mutations in the AT-rich interactive domain-containing protein 1A (ARID1A) have been reported to be associated with MSI through the inhibition of the protein binding with MSH2 [7].

Normally, when tumors are present in the body, they are phagocytosed by natural killer (NK) cells and macrophages, causing an immune response. The dendritic cells then present antigens, and the T cells attack tumor cells to prevent carcinogenesis. However, tumor cells can escape from the immune system and continue to grow. PD-L1 is expressed by the tumor cells, and the PD-1 expressed by the T cells engages with these, resulting in immunosuppression [8]. MSI-H tumors have many mutations, an increased risk of abnormal protein expression, and increased tumor immunogenicity. All of these are related to the efficacy of the immune checkpoint inhibitors against MSI-H tumors [9,10,11].

The frequency of MSI-H in ovarian cancer is reportedly 2–20%, and it is common mainly in endometrioid and clear cell carcinomas [12,13,14,15]. Ovarian endometrioid and clear cell carcinomas are called endometriosis-related ovarian neoplasms (ERONs), and endometriosis has been reported to be a cause of carcinogenesis. The phosphatase and tensin homolog deleted on chromosome10 (*PTEN)* deletion and microsatellite instability found in ERONs can also be confirmed in the epithelium of endometriosis [16,17,18]. Previously, we reported a high frequency of ARID1A mutations and suggested that this contributed to carcinogenesis in the ERONs in the Japanese population [19].

To date, while there have been some reports that utilized immunostaining of the MMR protein to determine the MSI status rather than an MSI analysis with a polymerase chain reaction (PCR), to the best of our knowledge there has been no report that evaluates the MSI status using a PCR, and focuses on ovarian endometrioid and clear cell carcinomas as a unique subset of ovarian carcinomas. In this study, ovarian endometrioid and clear cell carcinomas tissue samples were analyzed for their MSI status using a PCR, and the relationship between MSI-H, immune checkpoint molecules, and clinicopathological factors, including patient survival, was assessed.

## 2. Materials and Methods

### 2.1. Ethics Statement

This study was conducted in accordance with the ethical standards of national and international guidelines, as well as the Declaration of Helsinki. It was also approved by the Review Committee of the Shimane University Hospital. Tumor specimens were collected after obtaining written consent from all the patients with the approval of the Facility Ethical Committee (Shimane University Hospital, Izumo, Japan; approval No. 2004-0381, 5 March 2007).

### 2.2. Tissue Samples

This study evaluated 91 cases of ovarian carcinoma (52 clear cell carcinomas, 39 endometrioid carcinomas), which had received treatment between April 2008 and December 2018 at the Department of Obstetrics and Gynecology at the Shimane University Hospital and the Seirei Hamamatsu General Hospital. The tissues were formalin-fixed, paraffin-embedded tissue blocks. A pathologist diagnosed a sample stained with hematoxylin and eosin (Figure 1a,b). Ovarian cancer was staged according to the 2014 International Federation of Gynecology and Obstetrics (FIGO) guidelines [20]. The histological diagnoses were made in accordance with the 2014 World Health Organization (WHO) classification of ovarian cancer [21]. Clinical data were collected from the patient charts. The follow-up period ranged from 4 to 128 months, with a median follow-up of 57 months.

### 2.3. Immunohistochemistry

We evaluated the expression of the MMR proteins (MSH2, MLH1, MSH6, and PMS2), immune checkpoint molecules (PD-1 and PD-L1), CD8 lymphocyte infiltration in the tumor, and the ARID1A using immunohistochemistry (IHC) (Figure 2a–h and Figure 3a–h). The tissues were formalin-fixed, paraffin-embedded tissue blocks. They were sliced from the block to a thickness of 4 μm. We used antibodies against the MutS protein homolog 2 (MSH2, 1:50; Dako, Santa Clara, CA, USA), MutL protein homolog 1 (MLH1, 1:50; Dako), MutS protein homolog 6 (MSH6, 1:40; Dako), PMS1 homolog 2 (PMS2, 1:40; Dako), mouse monoclonal antibody [NAT105] to PD1 (NAT105, 1:100; Roche, Basel, Switzerland), rabbit monoclonal to PD-L1 (SP263, 1:100; Roche), and monoclonal mouse anti-human CD8 (SP57, 1:100; Dako).

A diagnosis of a deficient MMR (d-MMR) tumor was made if at least one of the four MMR proteins were negative; all other cases were considered to involve proficient MMR (p-MMR). We classified the expression of CD8 lymphocytes that infiltrated the tumor into four levels (0, undetectable; 1+, low density; 2+, moderate density; and 3+, high density). We considered as positive those grouped into the 2+ and 3+ groups. When 5% or more of the tumor-infiltrating lymphocytes were stained (membranous and cytoplasmic staining), they were considered to be PD-1-positive. Similarly, a PD-1-positive diagnosis was made when 5% or more of the tumors were stained. Two gynecologists (Y.N. and K.N.), who were blinded to the clinicopathological factors, assessed the experimental slides under a microscope.

### 2.4. Microsatellite Instability Analysis

All 91 cases were outsourced to BML (BML, Tokyo, Japan), and MSI was diagnosed using PCR. Genomic DNA was extracted from the paraffin slides of the tumors, and the number of repetitions of the five microsatellite markers (BAT25, BAT26, MONO27, NR21, and NR24) was evaluated using PCR. This method has been used in at least three of the five markers of the pentaplex assay, and enabling a conclusion to be reached without the need to analyze matching normal DNA [22].

### 2.5. Statistical Analysis

We divided the cases into the MSI-H and microsatellite stable (MSS) groups, and their associations with each clinicopathological and immune checkpoint factor was evaluated using the chi-squared test. In addition, we performed univariate analyses for progression-free survival (PFS) and overall survival (OS). PFS was defined as the period from the first treatment date to the recurrence date or the last follow-up date. OS was defined as the period from the first treatment to the death date or the last follow-up date. A survival analysis was performed for the PFS and OS and expressed as a Kaplan–Meier curve. The log-rank test was used to determine the statistical significance. In this study, we defined *p*-values below 0.05 as statistically significant. IBM SPSS Statistics for Windows, version 27 (IBM Corp., Armonk, NY, USA) was used for the data analysis.

## 3. Results

### 3.1. Clinicopathological Factors

We examined 52 clear cell carcinomas and 39 endometrioid carcinomas, retrospectively. The FIGO stages I and II were identified in 72 cases (43 clear cell carcinomas and 29 endometrioid carcinomas) while 19 cases were FIGO stage III/IV (9 clear cell carcinomas and 10 endometrioid carcinomas). Surgery was performed as the first treatment in 90 patients. Eighty patients had no residual tumors after the primary debulking surgery (PDS). In almost all cases, 3–6 courses of TC therapy (paclitaxel 175 mg/m^2^ and carboplatin area under the curve = 5 mg) were administered as postoperative chemotherapy. In addition, some cases using CPT-11 and cases in which intraperitoneal administration of carboplatin was added at the time of surgery, were also included. The clinicopathological factors of the patients are shown in Table 1.

### 3.2. Microsatellite Instability

In this study, we examined the MSI status using PCR. We also performed IHC for the MMR proteins (MSH2, MSH6, PMS2, and MLH1). From the PCR results, 5 of 91 (5%) cases were MSI-H, and all of these were endometrioid carcinomas. The number of MSI-H tumors was significantly higher in endometrioid carcinomas than in clear cell carcinomas (Table 1). The MSI-H group had an increased number of nucleotide sequence repetitions with two or more markers, including NR21 and BAT26 (Figure 4). Furthermore, in these five MSI-H cases, no correlations were found between the MSI-H and clinicopathological factors including age, FIGO stage, initial treatment, status of residual tumor, status of ARID1A expression, and the status of endometriosis (Table 1). In four of the five (80%) MSI-H cases, the MMR protein IHC results were negative. (Table 1). This was comparable to previous reports on the consistency of the findings between IHC and the MSI analyses [23].

### 3.3. Relationship between MSI-H and the Expression of CD8, PD-1, and PD-L1

Next, we examined the relationship between MSI-H and CD-8, PD-1, and PD-L1. In the MSI-H group, the expression of CD-8 was significantly higher (*p* = 0.026), while there were no significant differences in the expressions of PD-1 (*p* = 0.251) and PD-L1 (*p* = 0.664) (Table 1).

### 3.4. Prognostic Analysis Using the Kaplan–Meier Method

Survival curves were created for the PFS and OS of patients with MSI-H and MSS (Figure 5a). Neither group showed significant differences; however, in the ovarian endometrioid and clear cell carcinomas, the PFS tended to be better in the MSI-H group. In addition, a similar analysis was performed for the presence or absence of the expression of CD-8, PD-1, PD-L1, and MMR proteins. The OS was shorter in patients with a PD-L1 expression than in those without a PD-L1 expression (*p* = 0.022) (Figure 5b). There were no significant differences in the expression status of CD8, PD-1, MMR, and ARID1A (Appendix A).

## 4. Discussion

With a 5-year survival rate of 60%, ovarian cancer is the most common gynecological malignancy [2,3]. Moreover, since current treatments do not improve survival [24], a new treatment strategy is needed. Recently, with the development of genomic medicine, molecular-targeted therapies have attracted attention. The use of immune checkpoint inhibitors is being considered for use in MSI-H tumors. The frequency of MSI-H in ovarian carcinomas has been found to be 2–20% [12,13,14,15], in clear cell carcinoma 2.4–14.3%, and in endometrioid carcinoma 13.8–33% [13,14,15]. In our study, MSI-H was observed in 5% of all endometrioid and clear cell carcinomas, and MSI-H in approximately 13% of endometrioid carcinomas. These results are consistent with those of previous reports [13,14]. Howitt et al. reported that clear cell carcinomas display more MSI-H than endometrioid carcinomas [15]. However, in the present study, there were no cases of MSI-H in clear cell carcinomas. This disparity suggests that the genetic background of clear cell carcinomas may differ between Japanese and Western patients. Previously, we reported that the carcinogenic process may differ between Japanese and Westerners in low-grade serous carcinoma [25]. Our findings, taken together with those of our previous study and another report, suggest that collecting genetic data from more Japanese cases is necessary to clarify this issue.

d-MMR is considered to be caused by the mutation or promoter methylation of MMR-related genes; in fact, 80% (4/5) of the patients with MSI-H in this study also showed a loss of MMR protein. In contrast, only one MSI-H case with MMR protein expression showed a loss of ARID1A protein. This may support the report by Shen et al. that *ARID1A* gene mutations are involved in MSI-H without mutations or methylation of MMR-related genes [7].

There have been reports that the effectiveness of immune checkpoint inhibitors for MSI-H has increased immunogenicity [26]. MSI-H tumors display increased gene mutations and tumor mutational burden (TMB). Immune checkpoint inhibitors have been reported to be effective against TMB-high tumors [26]. Furthermore, immune checkpoint inhibitors are effective in cases with a high infiltration of CD-8-positive lymphocytes and cases with a high PD-L1 expression [27,28,29,30,31,32]. By inhibiting the PD-1 and PD-L1 pathways, tumor overgrowth may be suppressed. In our study, we found that the infiltration of CD-8 into tumors was significantly higher in the MSI-H group, confirming that the immune response was enhanced in MSI-H ovarian endometrioid carcinomas. However, we did not observe significant correlations between MSI-H and PD-1 or PD-L1 expressions, suggesting that the effects of the immune checkpoint inhibitors in ovarian endometrioid carcinomas may be limited. Ngheim et al. reported, in a closer view of our current findings, that PD-L1 expression is not always proportional to the effectiveness of immune checkpoint inhibitors [33].

There have been many reports that MSI-H tumors have a better prognosis than MSI-low and MSS in colorectal and gastric cancers [34,35,36]. MSI-H tumors have been shown to be more immunogenic, have better anti-tumor immune responses, and are more capable of inhibiting tumor cell growth [35,37,38,39]. It was discovered that the lymphocytes that invade the tumor are composed primarily of cytotoxic T lymphocytes, triggering a more specific anti-tumor immune response in rectal cancer [40]. Here, in the survival curve, no statistical correlations were found between the MSI-H group and the MSS group. On the other hand, PD-L1 expression is significantly associated with shorter OS in ovarian endometrioid and clear cell carcinomas. This prognostic difference between colorectal and ovarian cancers may be due to the relatively small number of analyzed cases and low prevalence of MSI-H tumors in ovarian endometrioid and clear cell carcinomas or in an organ-specific manner. Since immune checkpoint inhibitors are expected to be effective in tumors with a high PD-L1 expression [41], the positive PD-L1 cases with poor survival found in the current study may be aided using immune checkpoint inhibitors.

The main limitation of this study was the small number of cases that was analyzed. Further examination using an increased number of cases of Japanese ovarian endometrioid and clear cell carcinomas is necessary.

Some clinical studies have reported the efficacy and safety of nivolumab for platinum-resistant ovarian cancers, and a phase II study (KEYNOTE100) is currently underway. According to the reported studies, the disease control rate was relatively high at approximately 44–47.6% [41,42,43] and the OS was 17.6–18.7 months. Among patients with higher PD-L1 expressions, the OS was between 21.9 and 24.0 months. In addition, Kato et al. reported that, in an experimental model, the antitumor activity of lenvatinib plus anti PD-1 (immune checkpoint inhibitor) was greater than that of either single treatment [44]. Lenvatinib exhibits immunomodulatory activity by reducing the number of tumor-related macrophages. Combination therapy with immune checkpoint inhibitors induces potent antitumor activities through the activation of interferon (IFN) signaling pathways in the tumor immune microenvironments. Lwin et al. investigated the efficacy and safety of pembrolizumab and lenvatinib in combination therapy for advanced solid tumors, including triple-negative breast cancer and ovarian cancer (LEAP-005) in a phase II study in patients aged >18 years, with advanced cancer [45]. This trial included 31 patients with advanced ovarian cancers, and the disease control rate reached 74%, showing a significant improvement. While side effects (immune-related adverse events (irAEs) caused by the immune checkpoint inhibitors have been observed, they have been controllable, and strict control may lead to an improved prognosis of ovarian cancer [45]. Due to the low prevalence of MSI-H in ovarian endometrioid and clear cell carcinomas, the use of immune checkpoint inhibitors alone did not show a good response, while the use of combination therapy such as lenvatinib and immune checkpoint inhibitors may have had some effect. The effectiveness of combined treatment with immune checkpoint inhibitors and other agents, and the utility of biomarkers as indicators for the use of immune checkpoint inhibitors, have been reported with regard to ovarian cancers [46,47].

Previous studies have reported that treatment with platinum increases the PD-L1 expression in tumors. The treatment is useful in combination with other carcinoma immune checkpoint inhibitors [48,49] and the prognosis has shown an improvement with the addition of an immune checkpoint inhibitor to the standard treatment [50]. Furthermore, the response rate was increased with the use of a combination of checkpoint inhibitors [51]. Thus, it can be expected that the prognosis of ovarian endometrioid and clear cell carcinomas may have improved after the use of standard chemotherapy. Quite recently, it was reported that a combination of pembrolizumab and lenvatinib improved the prognosis of patients with endometrial cancers [52]. The authors of that study evaluated the prognosis of patients treated with platinum-combined chemotherapy for untreatable endometrial cancer by dividing them into a lenvatinib plus pembrolizumab group and a doxorubicin or paclitaxel monotherapy group. The median PFS was 7.2 and 3.8 months (HR = 0.56, *p* < 0.0001), respectively; this indicated a significant improvement in the lenvatinib plus pembrolizumab group. Although the study involved a different type of carcinoma, it demonstrated an improvement in the prognosis of patients who received concomitant immune checkpoint inhibitor treatment. Thus, there is a possibility that this treatment can improve the prognosis of patients with endometriosis-related ovarian cancer.

## 5. Conclusions

In conclusion, MSI-H is a very rare event in ovarian endometrioid and clear cell carcinomas. Furthermore, unlike other carcinomas, MSI-H is not a favorable prognostic factor in ovarian endometrioid and clear cell carcinomas. Therefore, improving the prognosis of these unique subtypes of ovarian carcinoma may require the use of a combination of immune checkpoint inhibitors and other molecular targeted therapies such as lenvatinib.

## Figures and Tables

**Figure 1 healthcare-10-00694-f001:**
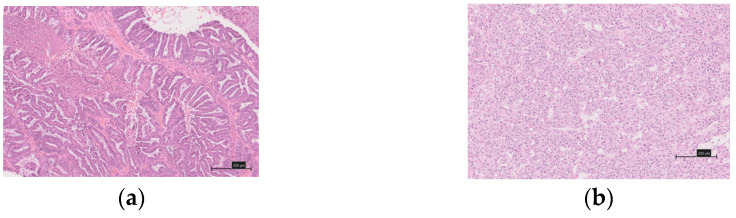
Hematoxylin and eosin (H&E) staining of ovarian endometrioid and clear cell carcinomas. (**a**) Representative H&E section showed ovarian endometrioid carcinoma. (**b**) Representative H&E section showed ovarian clear cell carcinoma.

**Figure 2 healthcare-10-00694-f002:**
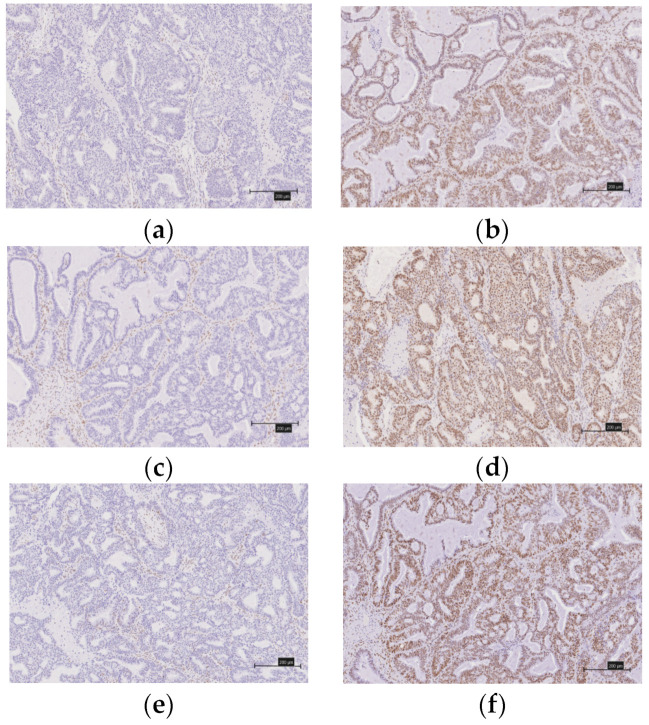
Immunostaining of mismatch repair proteins in ovarian endometrioid carcinomas. (**a**) There is no expression of MSH2. (**b**) There is expression of MSH2. (**c**) There is no expression of MLH1. (**d**) There is expression of MLH1. (**e**) There is no expression of MSH6. (**f**) There is expression of MSH6. (**g**) There is no expression of PMS2. (**h**) There is expression of PMS2.

**Figure 3 healthcare-10-00694-f003:**
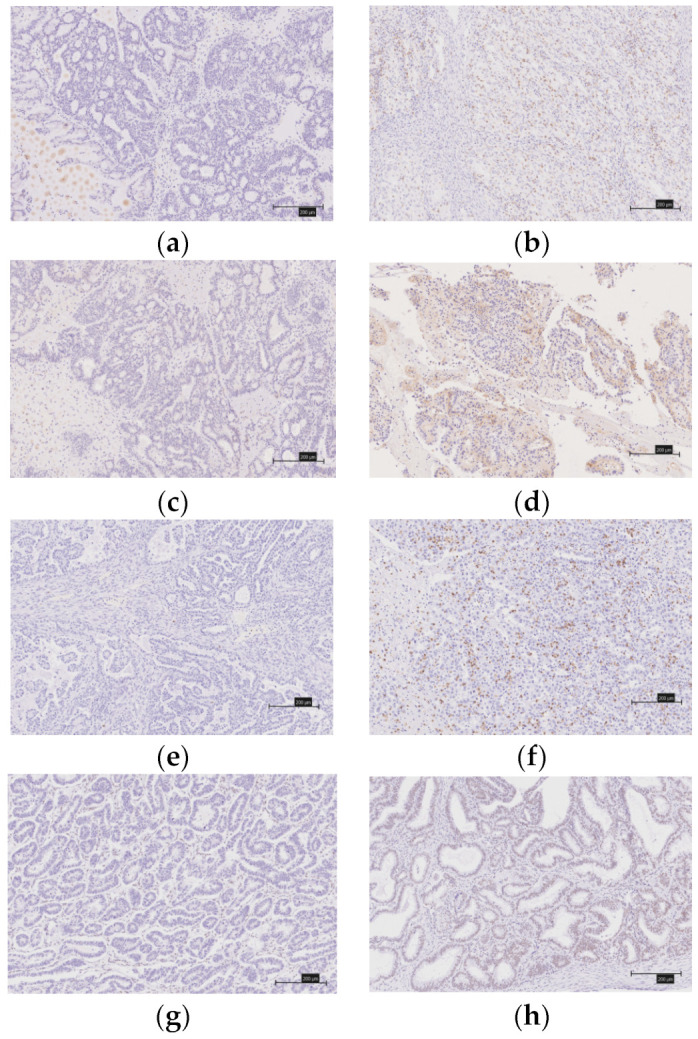
Immunostaining of immune checkpoint molecules (PD-1, PD-L1), CD8 lymphocytes infiltrations into the tumor and ARID1A. (**a**) There is no expression of PD-1. (**b**) There is expression of PD-1. (**c**) There is no expression PD-L1. (**d**) There is expression PD-L1. (**e**) CD8 expression score of 0. (**f**) CD8 expression score of 3+. (**g**) There is no expression of ARID1A. (**h**) There is expression of ARID1A.

**Figure 4 healthcare-10-00694-f004:**
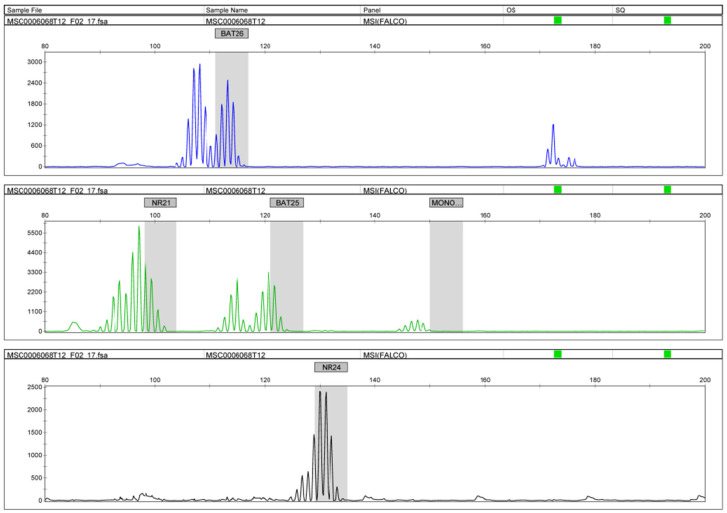
MSI analysis using PCR. Increases in the number of nucleotide sequence repetitions with BAT26 and NR21.

**Figure 5 healthcare-10-00694-f005:**
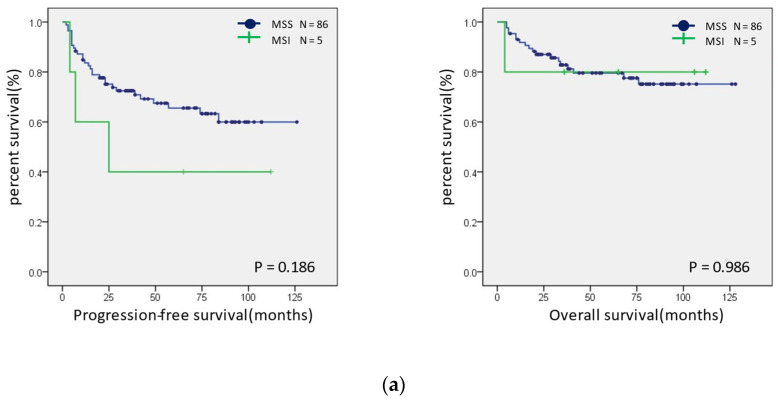
Prognostic analysis using Kaplan–Meier curves. (**a**) Progression-free (left panel) and overall (right panel) prognostic analysis of the MSI status in ovarian endometrioid and clear cell carcinomas. (**b**) Progression-free (left panel) and overall (right panel) prognostic analysis of the PD-L1 expression status in ovarian endometrioid and clear cell carcinomas.

**Table 1 healthcare-10-00694-t001:** Characteristics of ovarian cancer patients.

Characteristic	MSI-H	MSS	*p*-Value
	*N* = 5	*N* = 86	
**Age**			0.249
<60 years	4 (80%)	46 (53%)	
≥60 years	1 (20%)	40 (47%)	
**FIGO stage**			0.301
Ⅰ, Ⅱ	5 (100%)	67 (78%)	
Ⅲ, Ⅳ	0 (0%)	19 (22%)	
**Initial treatment**			0.055
PDS	4 (80%)	86 (100%)	
NAC	1 (20%)	0 (0%)	
**Residual tumor after PDS or IDS**			0.517
No residual tumor (R0)	5 (100%)	75 (87%)	
Yes	0 (0%)	11 (13%)	
**ARID1A**			0.406
Intact	2 (40%)	48 (56%)	
Loss	3 (60%)	38 (44%)	
**MMR protein (IHC)**			0.000
p-MMR	1 (20%)	80 (93%)	
d-MMR	4 (80%)	6 (7%)	
**Endometriosis**			0.265
No	4 (80%)	47 (55%)	
Yes	1 (20%)	39 (45%)	
**Organization type**			0.012
Clear cell	0 (0%)	52 (60%)	
Endometrioid	5 (100%)	34 (40%)	
**CD-8**			0.026
Positive	4 (80%)	23 (27%)	
Negative	1 (20%)	63 (73%)	
**PD-1**			0.251
Positive	1 (20%)	4 (5%)	
Negative	4 (80%)	82 (95%)	
**PD-L1**			0.664
Positive	0 (0%)	7 (8%)	
Negative	5 (100%)	79 (92%)	

All data are expressed as number (%). MSI: microsatellite instability; MSS: microsatellite stable; FIGO: International Federation of Gynecology and Obstetrics; PDS: primary debulking surgery; NAC: neoadjuvant chemotherapy; IDS: interval debulking surgery; MMR: mismatch repair; IHC: immunohistochemistry; p-MMR: proficient MMR; d-MMR: deficient MMR.

## Data Availability

The data presented in this study are available on request from the corresponding author (K.N.).

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
