# Peer review of "Ovarian Endometrioid and Clear Cell Carcinomas with Low Prevalence of Microsatellite Instability: A Unique Subset of Ovarian Carcinomas Could Benefit from Combination Therapy with Immune Checkpoint Inhibitors and Other Anticancer Agents"

_healthcare, 2022, doi:10.3390/healthcare10040694_

Round 1
Reviewer 1 Report
This retrospective study aims to analyze a specific subtype of epithelial ovarian carcinoma (clear cells and endometrioid) evaluating the presence of microsatellite instability (MSI) and correlating this data to clinical pathological factors, including survival analysis.
Specific comments:
Title should be changed : the title of the article clearly refers to a specific treatment option for the subgroup of ovarian cancer analyzed. However, no data relating to the treatment are specified, as this is a retrospective analysis in which it is assumed that no case underwent therapy under investigation. Therefore it is a descriptive analysis of the population which, with few significant data, does not well refute the immunotherapy treatment option that is exclusively desumed from the analysis of other studies.
Lines 65-66 : can more recent bibliographic references be used?
Paragraph 3.1 Clinicopathological factors: put all the percentages to make the data clearer and more immediate. Data related treatment are summarized in the text, and are not specified in the table. Table 1 about the section in question has some inaccuracies that do not make it immediate to read; the total number and its percentage is not clearly expressed in the description (N= must be a number not a percentage), and it is not specified that the numbers in parentheses are percentages. Missing data about treatment.
Line 189: “endometrial carcinomas” is a mistake, please write “endometrioid”.
Lines 203-204: I would underline that the data relating to the molecular of PD-1 and PD-L1 is negative, then secondarily specifying its non-significance. Review table 2 as suggested for table 1.
Line 228: the reference 24 is missing and probably from that point on all the other references are mismatched, please verify!
The LEAP 005 study cited obtained the most encouraging results in the non-MSI-H / pMMR subpopulation. This data, not clearly expressed in the discussion, is not in line with the conclusions of the study; or else the conclusion should be better expressed. Furthermore, with reference to the data relating to the use of treatment combinations (eg chemotherapy + immunotherapy) I would use more recent and more consistent data with other ovarian cancer studies.
Check the References because do not match.
Author Response
Point-wise responses to the comments made by Reviewer 1:
Comments and Suggestions for Authors
Point 1)
Title should be changed : the title of the article clearly refers to a specific treatment option for the subgroup of ovarian cancer analyzed. However, no data relating to the treatment are specified, as this is a retrospective analysis in which it is assumed that no case underwent therapy under investigation. Therefore it is a descriptive analysis of the population which, with few significant data, does not well refute the immunotherapy treatment option that is exclusively desumed from the analysis of other studies.
Answer 1)
Thank you for your valuable insights and comments. We have modified the title to “Ovarian endometrioid and clear cell carcinomas with low prevalence of microsatellite instability: A unique subset of ovarian carcinomas could benefit from combination therapy with immune checkpoint inhibitors and other anticancer agents”. Please check if this is more appropriate. If you have any further suggestions, please let us know, and we will be happy to incorporate them.
Point 2)
Lines 65-66 : can more recent bibliographic references be used?
Answer 2)
Thank you for your advice. We have revised the citations to more recent ones (references 12, 13, 16).
Point 3)
Paragraph 3.1 Clinicopathological factors: put all the percentages to make the data clearer and more immediate. Data related treatment are summarized in the text, and are not specified in the table. Table 1 about the section in question has some inaccuracies that do not make it immediate to read; the total number and its percentage is not clearly expressed in the description (N= must be a number not a percentage), and it is not specified that the numbers in parentheses are percentages. Missing data about treatment.
Answer 3)
We apologize for the lack of clarity. We have modified Table 1 to facilitate better understanding. In addition, we have added the following sentence in the footnote: “The data are expressed as number (%).”
Point 4)
Line 189: “endometrial carcinomas” is a mistake, please write “endometrioid”.
Answer 4)
We apologize for the overlook. The term has been revised to “endometrioid”. (line 241).
Point 5)
Lines 203-204: I would underline that the data relating to the molecular of PD-1 and PD-L1 is negative, then secondarily specifying its non-significance. Review table 2 as suggested for table 1.
Answer 5)
Thank you for your recommendations. Tables 1 and 2 are now combined into a single Table 1.
Point 6)
Line 228: the reference 24 is missing and probably from that point on all the other references are mismatched, please verify!
Answer 6)
Thank you for pointing this out. We have revised the citations in the revised manuscript.
Point 7)
The LEAP 005 study cited obtained the most encouraging results in the non-MSI-H / pMMR subpopulation. This data, not clearly expressed in the discussion, is not in line with the conclusions of the study; or else the conclusion should be better expressed. Furthermore, with reference to the data relating to the use of treatment combinations (eg chemotherapy + immunotherapy)
I would use more recent and more consistent data with other ovarian cancer studies.
Answer 7) Thank you for your valuable comments and suggestions. We have made some revisions based on the results of the Keynote 775 study. We have also discussed our findings with reference to recent ovarian cancer studies (lines 387-391 and 401-414).
Reviewer 2 Report
In this study, the authors evaluated MSI using PCR on ovarian endometrioid and clear cell carcinomas. They found five MSI-H cases with higher CD-8 expression and that patients with PD-L1 expression had shorter overall survival than those without. The novelty of the study is to use PCR to determine MSI. However, the study has several limitations, and my comments are listed below:
- I am confused by the definition of MSI and MSI-H. Is MSI and MSI-H the same here? It seems that the authors used the two terms randomly. Please correct it.
- In line 140, “A diagnosis of a MSI-H tumor is made if at least one of the four MMR proteins are negative”. In line 186, “In this study, we examined the MSI using PCR. We also performed IHC for the MMR proteins (MSH2, MSH6, PMS2, and MLH1).” Which feature was used to define MSI-H (PCR or IHC)? Table shows the MSI result based on PCR. How about IHC for Table 1 (also for Table 2 and Figure 5)?
- A comparison between PCR and IHC-based identification of MSI should be made in results and discussion sections. What’s the purpose or benefit of using PCR instead of IHC?
- The study cohort have a relatively low frequency of MSI cases for endometrioid carcinoma compared to other studies (12.8 % vs 13.8–33%). This may result from my question#2 related to the definition of MSI.
- It is known that MSI occurred in endometrioid carcinoma more frequently than clear cell carcinomas. Please justify why included clear cell carcinoma for the investigation of MSI and why included more clear cell carcinomas than endometrioid carcinoma. The inclusion rationale impacts the result of the study because only five MSI cases were identified. Five cases are difficult to make any conclusion for Table 2 and Figure 5.
Author Response
Point-wise response to the comment made by Reviewer 2:
Comments and Suggestions for Authors
Point 1)
I am confused by the definition of MSI and MSI-H. Is MSI and MSI-H the same here? It seems that the authors used the two terms randomly. Please correct it.
Answer 1)
We apologize for the lack of clarity. MSI-H was considered when diagnosis was made using PCR, whereas deficient MMR was considered when diagnosis was based on MMR protein deficiency in IHC. We have revised the manuscript accordingly.
Point 2)
In line 140, “A diagnosis of a MSI-H tumor is made if at least one of the four MMR proteins are negative”. In line 186, “In this study, we examined the MSI using PCR. We also performed IHC for the MMR proteins (MSH2, MSH6, PMS2, and MLH1).” Which feature was used to define MSI-H (PCR or IHC)? Table shows the MSI result based on PCR. How about IHC for Table 1 (also for Table 2 and Figure 5)?
Answer 2)
Thank you for pointing this out. MSI-H was considered when diagnosis was made using PCR, whereas deficient MMR was considered when diagnosis was based on MMR protein deficiency in IHC. We have revised the manuscript accordingly (lines 175-177). We have also combined Tables 1 and 2 into a single Table 1, which includes IHC data.
Point 3)
A comparison between PCR and IHC-based identification of MSI should be made in results and discussion sections. What’s the purpose or benefit of using PCR instead of IHC?
Answer 3)
Currently, in clinical settings, MSI is diagnosed using PCR. Therefore, PCR was used in the current study.
Point 4)
The study cohort have a relatively low frequency of MSI cases for endometrioid carcinoma compared to other studies (12.8 % vs 13.8–33%). This may result from my question#2 related to the definition of MSI.
Answer 4)
Thank you for your valuable comments and suggestions.
Point 5)
It is known that MSI occurred in endometrioid carcinoma more frequently than clear cell carcinomas. Please justify why included clear cell carcinoma for the investigation of MSI and why included more clear cell carcinomas than endometrioid carcinoma. The inclusion rationale impacts the result of the study because only five MSI cases were identified. Five cases are difficult to make any conclusion for Table 2 and Figure 5.
Answer 5)
Thank you for your comments. In this study, we focused on endometriosis-related ovarian neoplasms (ERONs). Therefore, endometrioid carcinoma and clear cell carcinoma were selected. Because the frequency of clear cell carcinomas is higher than that of endometrioid carcinomas in Japan (S.B. Coburn et. al. International patterns and treands in ovarian cancer incidence, overall and by histologic subtype. Cancer Epidermiology. 2017 March.), more clear cell carcinomas than endometrioid carcinomas were included in this study, and we were able to show results that are more suitable for Japanese populations. However, as you pointed out, the number of cases of MSI-H was small, so accumulation of more data is necessary in the future.
Round 2
Reviewer 2 Report
The authors have answered most of my questions.